# Three-Dimensional Stress Fields in Thick Orthotropic Plates with Sharply Curved Notches under In-Plane and Out-of-Plane Shear

**DOI:** 10.3390/polym15092013

**Published:** 2023-04-24

**Authors:** Alessandro Pontefisso, Matteo Pastrello, Michele Zappalorto

**Affiliations:** Department of Management and Engineering, University of Padova, Stradella San Nicola 3, 36100 Vicenza, Italy

**Keywords:** three-dimensional anisotropic elasticity, blunt crack, stress field

## Abstract

In this paper, an analytical solution for the stress fields in the close neighbourhoods of radiused notches in thick orthotropic plates under shear loading and twisting is provided. In the first step, the equations of the three-dimensional theory of elasticity are successfully reduced to two uncoupled equations in two-dimensional space. Later, the 3D stress field solution for orthotropic plates with radiused notches is presented and its degree of accuracy is discussed by comparing theoretical results and numerical data from 3D FE analyses. The solution proposed can be satisfactorily used to characterise the stress field in plates made with polymeric composite materials, such as fibre-reinforced polymers and natural composites.

## 1. Introduction

The regions close to geometrical variations, i.e., notches, holes and cutouts, are characterized by a significant concentration of the stress fields, which are responsible for crack formation under static and cyclic loading conditions. 

Predicting the fatigue crack nucleation or the static failure of notched solids is of primary concern for mechanical designers, and the formulation of effective strength criteria stems from the accurate knowledge of the local stress fields, thus justifying the significant attention paid by different researchers on this topic over the years (see, among the others, Refs. [1,2,3,4,5,6,7,8,9,10,11,12] and references quoted therein).

All the above-mentioned papers refer to two-dimensional analyses, even if the mechanical components are three-dimensional by nature, and the number of analytical works in the literature devoted to 3D stress fields is very limited, especially in the case of orthotropic materials. 

Dealing with this topic, it is worth mentioning the scientific results by Prabhu and Lambros published in 2000 [13], where a numerical investigation on the 3D stress distribution in a cracked orthotropic plate is carried out, and by Cheng et al. [14] who, in 2015, studied the V-notch problem in an orthotropic solid under a generalised plane deformation state. Differently, Pageau and Biggers in 1996 [15] developed a 3D finite element formulation to derive the singular stress state near material and geometric discontinuities in orthotropic and anisotropic media, whereas Choi and Folias in 1994 [16] analysed the three-dimensional stress fields in a composite plate weakened by a hole.

On parallel tracks, in the early 2000s, Kotousov and Wang [17,18] studied transversely isotropic composite plates with holes or inclusions, proposing a generalised plane-strain theory, whilst later on, Zappalorto and Carraro studied thick anisotropic plates with pointed V-notches [19,20], and in 2017, developed a new three-dimensional theory for anisotropic blunt cracks [21]. 

Moving to recent progresses in the field, worthy of mention are the results of Shi et al. [22], who developed a method for designing thin cylindrical shells for aerospace applications where, in the presence of deep single-side notches, the stress concentration factor tends to zero. The results were validated with FE analyses and a series of experimental activities, also using an indirect method for measuring notch stresses [23]. 

According to Zappalorto and co-workers [19,20,21], the full 3D stress field problem can be simplified into two uncoupled equations, each defined in a bi-dimensional domain; the former refers to the solution of the corresponding plane notch problem, while the latter refers to the solution of the corresponding out-of-plane shear notch problem. The results reported in Refs. [19,20,21] highlight the existence, also in the case of orthotropic and anisotropic bodies, of coupled modes already detected in thick isotropic plates [24,25,26,27,28,29].

The present paper’s goals can be listed as follows: To provide evidence that the 3D solution derived by Zappalorto and Carraro [19] for pointed notches can be extended also to orthotropic plates with holes or lateral radiused notches with any notch opening angle, under the hypothesis of a sufficiently small notch tip radius;To show that, on the basis of the plane solution, stress components σ_xx_, σ_yy_ and τ_xy_ in a thick 3D anisotropic notched plate (i.e., the in-plane stress fields) can be accurately determined, whereas out-of-plane shear stresses, $\tau_{xz}$ and $\tau_{yz}$, can be assessed using the pure antiplane shear solution;To also document the presence of coupled modes for orthotropic thick plates weakened by holes or lateral notches with any notch opening angle, as reported in other research articles concerning isotropic components.

Finally, it is worth mentioning that the obtained solution is very useful to analyse the stress fields in polymeric composites, such as carbon-fibre-reinforced epoxy laminates, short fibre-reinforced polymers or natural composites.

## 2. Simplified Three-Dimensional Elasticity Theory for Orthotropic Thick Plates

### 2.1. Basic Field Equation

Consider a thick orthotropic plate weakened by a radiused notch. With reference to a Cartesian orthogonal system centred on the symmetry plane of the plate (see Figure 1 or Figure 2), the stress–strain relationships can be written as:(1)$$\left\{\begin{array}{c} \varepsilon_{xx}\\ \varepsilon_{yy}\\ \varepsilon_{zz}\\ \begin{array}{l} \gamma_{yz}\\ \gamma_{xz}\\ \gamma_{xy} \end{array} \end{array}\right\}=\left[\begin{array}{cccccc} S_{11} & S_{12} & S_{13} & 0 & 0 & 0\\ . & S_{22} & S_{23} & 0 & 0 & 0\\ . & . & S_{33} & 0 & 0 & 0\\ . & . & . & S_{44} & 0 & 0\\ . & . & . & . & S_{55} & 0\\ . & . & . & . & . & S_{66} \end{array}\right]\left\{\begin{array}{c} \sigma_{xx}\\ \sigma_{yy}\\ \sigma_{zz}\\ \begin{array}{l} \tau_{yz}\\ \tau_{xz}\\ \tau_{xy} \end{array} \end{array}\right\}$$
where S_ij_ are conventional compliance coefficients or, equivalently:(2)$$\left\{\begin{array}{c} \sigma_{xx}\\ \sigma_{yy}\\ \sigma_{zz}\\ \begin{array}{l} \tau_{yz}\\ \tau_{xz}\\ \tau_{xy} \end{array} \end{array}\right\}=\left[\begin{array}{cccccc} C_{11} & C_{12} & C_{13} & 0 & 0 & 0\\ . & C_{22} & C_{23} & 0 & 0 & 0\\ . & . & C_{33} & 0 & 0 & 0\\ . & . & . & C_{44} & 0 & 0\\ . & . & . & . & C_{55} & 0\\ . & . & . & . & . & C_{66} \end{array}\right]\left\{\begin{array}{c} \varepsilon_{xx}\\ \varepsilon_{yy}\\ \varepsilon_{zz}\\ \begin{array}{l} \gamma_{yz}\\ \gamma_{xz}\\ \gamma_{xy} \end{array} \end{array}\right\}$$
where C_ij_ are stiffness coefficients.

Suppose now that displacement fields near the notch tip could be written in the following form by means of separated variables [19,28]:(3)$$u_{x}=f(z)\times u(x,y)\;\;u_{y}=f(z)\times v(x,y)\;\;u_{z}=g\left(z\right)\times w(x,y)$$

Invoking strain definitions, one obtains:(4)$$\begin{array}{l} \varepsilon_{\mathrm{xx}}=\frac{\partial u_{x}}{\partial x}=f(z)\times \frac{\partial u(x,y)}{\partial x}\;\;\;\;\;\;\gamma_{\mathrm{xy}}=\frac{\partial u_{x}}{\partial y}+\frac{\partial u_{y}}{\partial x}=f(z)\times \left(\frac{\partial u(x,y)}{\partial y}+\frac{\partial v(x,y)}{\partial x}\right)\\ \varepsilon_{\mathrm{yy}}=\frac{\partial u_{y}}{\partial y}=f(z)\times \frac{\partial v(x,y)}{\partial y}\;\;\;\;\;\;\gamma_{\mathrm{xz}}=\frac{\partial u_{x}}{\partial z}+\frac{\partial u_{z}}{\partial x}=f^{\prime }(z)\times u(x,y)+g(z)\times \frac{\partial w(x,y)}{\partial x}\\ \varepsilon_{\mathrm{zz}}=\frac{\partial u_{z}}{\partial z}=g^{\prime }(z)\times w(x,y)\;\;\;\;\;\gamma_{\mathrm{yz}}=\frac{\partial u_{y}}{\partial z}+\frac{\partial u_{z}}{\partial y}=f^{\prime }(z)\times v(x,y)+g(z)\times \frac{\partial w(x,y)}{\partial y} \end{array}$$

Under the hypothesis that the tip radius of the radiused notch is small enough to assure that $f^{\prime }(z)\times u(x,y)$, $f^{\prime }(z)\times v(x,y)$ and $g^{\prime }(z)\times w(x,y)$ can be disregarded when compared to g(z)⋅∂w(x,y)/∂x and g(z)⋅∂w(x,y)/∂y, the strain field can be written as [19,20,21]:(5)$$\begin{array}{l} \varepsilon_{xx}=f(z)\times \frac{\partial u(x,y)}{\partial x}\;\;\;\gamma_{xy}=f(z)\times \left(\frac{\partial u(x,y)}{\partial y}+\frac{\partial v(x,y)}{\partial x}\right)\\ \varepsilon_{yy}=f(z)\times \frac{\partial v(x,y)}{\partial y}\;\;\;\gamma_{xz}≅g(z)\times \frac{\partial w(x,y)}{\partial x}\\ \varepsilon_{zz}≅0\;\;\;\;\;\gamma_{yz}≅g(z)\times \frac{\partial w(x,y)}{\partial y} \end{array}$$

As a result of Equation (5), a state of “quasi” plane strain is present in the stress concentration areas near the notch tip; indeed, the through-the-thickness normal strain can be disregarded, whereas the out-of-plane shear strains cannot. Equation (5) further gives:(6)$$\begin{array}{l} \sigma_{zz}≅-\;\frac{S_{13}\sigma_{xx}+S_{23}\sigma_{yy}}{S_{33}}≅f(z)\left\{C_{13}\frac{\partial u(x,y)}{\partial x}+C_{23}\frac{\partial v(x,y)}{\partial y}\right\}\\ \tau_{\mathrm{yz}}≅C_{44}g(z)\times \frac{\partial w(x,y)}{\partial y}\\ \tau_{\mathrm{xz}}≅C_{55}g(z)\times \frac{\partial w(x,y)}{\partial x} \end{array}$$

Moreover, Equation (5) combined with Equation (1) gives the possibility to formulate the stress components according to the following expressions:(7)$$\begin{array}{l} \sigma_{\mathrm{xx}}=f(z)\times {\tilde{\sigma }}_{\mathrm{xx}}\left(x,y\right)\;\;\sigma_{\mathrm{yy}}=f(z)\times {\tilde{\sigma }}_{\mathrm{yy}}\left(x,y\right)\;\;\sigma_{\mathrm{zz}}=f(z)\times {\tilde{\sigma }}_{\mathrm{zz}}\left(x,y\right)\\ \tau_{\mathrm{xy}}=f(z)\times {\tilde{\tau }}_{\mathrm{xy}}\left(x,y\right)\;\;\tau_{\mathrm{xz}}=g(z)\times {\tilde{\sigma }}_{\mathrm{xz}}\left(x,y\right)\;\;\tau_{\mathrm{yz}}=g(z)\times {\tilde{\sigma }}_{\mathrm{yz}}\left(x,y\right) \end{array}$$

Consider now the equilibrium equation in the z direction:(8)$$\frac{\partial \tau_{\mathrm{xz}}}{\partial^{}x}+\frac{\partial \tau_{\mathrm{yz}}}{\partial^{}y}+\frac{\partial \sigma_{\mathrm{zz}}}{\partial^{}z}=0$$

Substituting Equation (6) into Equation (8) results in:(9)$$\frac{\partial }{\partial^{}x}\left\{C_{55}g(z)\times \frac{\partial w(x,y)}{\partial x}+f^{\prime }(z)C_{13}\cdot u(x,y)\right\}+\frac{\partial }{\partial^{}y}\left\{C_{44}g(z)\times \frac{\partial w(x,y)}{\partial y}+f^{\prime }(z)C_{23}\cdot v(x,y)\right\}=0$$
and since $f^{\prime }(z)\times u(x,y)$ and $f^{\prime }(z)\times v(x,y)$ can be disregarded if compared to g(z)⋅∂w(x,y)/∂x and g(z)⋅∂w(x,y)/∂y, Equation (9) further simplifies to give:(10)$$C_{55}\frac{\partial^{2}w(x,y)}{\partial^{}x^{2}}+C_{44}\frac{\partial^{2}w(x,y)}{\partial^{}y^{2}}=0$$
where C_ij_ are stiffness coefficients, or, more explicitly:(11)$$G_{xz}\frac{\partial^{2}w}{\partial^{}x^{2}}+G_{yz}\frac{\partial^{2}w}{\partial^{}y^{2}}=0$$

Concerning the equilibrium equations in the x-direction and in the y-direction, the Airy stress function, ϕ, defined in the (x,y) plane, guarantees that such equilibrium is preserved [19,20,21]:(12)$${\tilde{\sigma }}_{xx}=\frac{\partial^{2}\phi }{\partial y^{2}}\;\;\;{\tilde{\sigma }}_{yy}=\frac{\partial^{2}\phi }{\partial x^{2}}\;\;\;{\tilde{\sigma }}_{xy}=-\frac{\partial^{2}\phi }{\partial x\partial y}$$

Substituting Equation (12) into Equation (1), further invoking the compatibility equation in the (x,y) plane gives the possibility to formulate the following fourth-order differential equation:(13)$$B_{22}\frac{\partial^{4}\phi }{\partial x^{4}}+(2B_{12}+B_{66})\frac{\partial^{4}\phi }{\partial x^{2}\partial y^{2}}+B_{11}\frac{\partial^{4}\phi }{\partial y^{4}}=0$$
where:(14)$$B_{ij}=S_{ij}-\frac{S_{i3}^{}S_{j3}^{}}{S_{33}}$$

Equations (10) and (13) must be both valid at the same time. Accordingly, by solving the following two differential equations (which are uncoupled) it is possible to simplify the equations that describe the three-dimensional elastic fields for a plate with a notch:{(15)B22∂4ϕ∂x4+(2B12+B66)∂4ϕ∂x2∂y2+B11∂4ϕ∂y4=0(16)C55∂2w∂x2+C44∂2w∂y2=0

One should note that Equation (15) is linked to the plane strain problem of the orthotropic theory of elasticity, whereas Equation (16) is the equation defining the solution of the antiplane one. 

The practical consequence of Equations (15) and Equation (16) is that the application of an out-of-plane shear stress, expressed by a non-zero w function, induces a local non-zero Airy ϕ function due to three-dimensional effects. At the same time, the application of any kind of in-plane loading (such as bending, tension or in-plane shear) generates a local non-zero w function. This coupled behaviour is not accounted for by the standard plane strain or plain stress solutions.

In particular, load conditions applied to a thick plate resulting in a skew-symmetric ϕ function (mode 2) create, due to three-dimensional effects, an antisymmetric w function (mode 3) in the close neighbourhood of the notch tip.

### 2.2. Solution for the Quasi-Biharmonic Equation (In-Plane Stress Field Components and Out-of-Plane Normal Stress σ_zz_)

In-plane stress fields in a thick notched orthotropic plate can be written in terms of two complex functions [7,10]:(17)σxx=Re{ μ12φ1(z1)+μ22φ2(z2) }σyy=Re{ φ1(z1)+φ2(z2) }τxy=−Re{ μ1φ1(z1)+μ2φ2(z2)  }
in cartesian coordinates, or: (18)σrr=2Re{ (sinθ−μ1cosθ)2φ1′(z1)+(sinθ−μ2cosθ)2φ2′(z2) }σθθ=2Re{ (cosθ+μ1sinθ)2φ1′(z1)+(cosθ+μ2sinθ)2φ2′(z2) }τrθ=2Re{ (sinθ−μ1cosθ)(cosθ+μ1sinθ)φ1′(z1)+(sinθ−μ2cosθ)(cosθ+μ2sinθ)φ2′(z2) }
in polar coordinates.

In Equations (17) and (18), μ_1_ and μ_2_ are unequal imaginary numbers defined as:(19)$$\mu_{1}=i\beta_{1}\;\;\;\;\mu_{2}=i\beta_{2}\;\;\;\;\;(\beta_{1},\;\beta_{2}>0)$$
and represent the roots of the following characteristic equation [7,10,30]:(20)$$B_{11}\mu_{}^{4}+(2B_{12}+B_{66})\mu_{}^{2}+B_{22}=0$$
under the condition that ${\left(2B_{12}+B_{66}\right)}^{2}\geq 4B_{11}B_{22}$.

Differently, z_1_ and z_2_ are complex variables which are defined as:(21)$$z_{1}=x+\mu_{1}y=r\rho_{1}\;e^{i\;\theta_{1}}\;\;\;z_{2}=x+\mu_{2}y=r\rho_{2}\;e^{i\;\theta_{2}}$$

Equation (18) can be conveniently re-written as [10]:(22)σrr=2Re{ (k11+i k12)φ1′(z1)+(k21+i k22)φ2′(z2) }σθθ=2Re{ (m11+i m12)φ1′(z1)+(m21+i m22)φ2′(z2) }τrθ=2Re{ (n11+i n12)φ1′(z1)+(n21+i n22)φ2′(z2) }
where the auxiliary angular functions introduced, k_ij_, m_ij_ and n_ij_, read as follows:(23)$$k_{11}=\sin^{2}\theta -{\left(\beta_{1}\cos\theta \right)}^{2}\;\;\;\;\;\;k_{12}=-2\beta_{1}\sin\theta \cos\theta$$
(24)$$k_{21}=\sin^{2}\theta -{\left(\beta_{2}\cos\theta \right)}^{2}\;\;\;\;\;\;k_{22}=-2\beta_{2}\cos\theta \sin\theta$$
(25)$$m_{11}=\cos^{2}\theta -{\left(\beta_{1}\sin\theta \right)}^{2}\;\;\;\;\;\;m_{12}=2\beta_{1}\sin\theta \cos\theta$$
(26)$$m_{21}=\cos^{2}\theta -{\left(\beta_{2}\sin\theta \right)}^{2}\;\;\;\;\;\;m_{22}=2\beta_{2}\sin\theta \cos\theta$$
(27)$$n_{11}=\frac{1}{2}\sin2\theta \left(1+\beta_{1}^{2}\right)\;\;\;\;\;\;n_{12}=-\beta_{1}\cos2\theta$$
(28)$$n_{21}=\frac{1}{2}\sin2\theta \left(1+\beta_{2}^{2}\right)\;\;\;\;\;\;n_{22}=-\beta_{2}\cos2\theta$$

### 2.3. Solution for the Quasi-Harmonic Equation (Out-of-Plane Shear Stresses)

Out-of-plane shear stress fields in the considered thick orthotropic plate can be written in terms of a complex function [31]:(29)τzx=Re{ μ3φ3′(z3)}     τzy=−Re{ φ3′(z3) }
in cartesian coordinates, or:(30)τzθ=−2Re{ (cosθ+i β3sinθ)φ3′(z3)}τzr=2Re{ (−sinθ+i β3cosθ)φ3′(z3) }
in polar coordinates, where φ_3_ is a proper complex function to be chosen depending on the specific notch geometry under consideration, z_3_ = x + μ_3_y, and:(31)$$\mu_{3}=\pm i\beta_{3}=\pm i\sqrt{\frac{g_{xz}}{g_{yz}}}$$

## 3. Elliptical Hole in a Thick Plate under Shear

Consider a thick plate with an elliptical hole under shear or torsion (see Figure 1).

The in-plane stress fields can be determined according to the solution proposed by Savin [32]. Re-arranging Savin equations allows the explicit solution to this problem to be obtained, according to which the stress field is:(32)$$\begin{array}{l} \sigma_{\mathrm{xx}}=\frac{\tau_{\mathrm{xy}}^{\mathrm{Max}}\left(z\right)}{\omega }\cdot \frac{\left({a+b\beta }_{1}\right)\left({a+b\beta }_{2}\right)\left(r_{1}\beta_{2}{}^{2}\Lambda_{2}\Theta_{1}{-r}_{2}\beta_{1}{}^{2}\Lambda_{1}\Theta_{2}\right)}{r_{1}r_{2}\left(\beta_{1}{-\beta }_{2}\right)\Theta_{1}\Theta_{2}}\\ \sigma_{\mathrm{yy}}=\frac{\tau_{\mathrm{xy}}^{\mathrm{Max}}\left(z\right)}{\omega }\cdot \frac{\left({a+b\beta }_{1}\right)\left({a+b\beta }_{2}\right)\left(r_{2}\Lambda_{1}\Theta_{2}{-r}_{1}\Lambda_{2}\Theta_{1}\right)}{r_{1}r_{2}\left(\beta_{1}{-\beta }_{2}\right)\Theta_{1}\Theta_{2}}\\ \tau_{\mathrm{xy}}=\frac{\tau_{\mathrm{xy}}^{\mathrm{Max}}\left(z\right)}{\omega }\left[1+\frac{\left({a+b\beta }_{1}\right)\left({a+b\beta }_{2}\right)\left(r_{2}\beta_{1}\Theta_{1}\Omega_{2}{-r}_{1}\beta_{2}\Theta_{2}\Omega_{1}\right)}{r_{1}r_{2}\left(\beta_{1}{-\beta }_{2}\right)\Theta_{1}\Theta_{2}}\right] \end{array}$$
where $\tau_{\mathrm{xy}}^{\mathrm{Max}}\left(z\right)$ is, for a given z value, the maximum shear stress occurring along the notch bisector line at a certain distance, x_Max_-a, from the notch tip, whereas:(33)$$\begin{array}{l} r_{i}=\sqrt[{4}]{{4x}^{2}y^{2}\beta_{i}^{2}+{\left(a^{2}{-x}^{2}+\left(y^{2}{-b}^{2}\right)\beta_{i}^{2}\right)}^{2}}\\ \theta_{i}=\mathrm{Arg}\left\{\left[\left(x^{2}{-a}^{2}\right){-\beta }_{i}^{2}\left(y^{2}{-b}^{2}\right)\right]+i\left[{-2\mathrm{xy}\beta }_{i}^{}\right]\right\} \end{array}$$
(34)$$\begin{array}{l} \Theta_{i}=x^{2}{+r}_{i}{}^{2}{+y}^{2}\beta_{i}^{2}{+2r}_{i}\left(\mathrm{xcos}\frac{\theta_{i}}{2}{+y\beta }_{i}\sin\frac{\theta_{i}}{2}\right)\\ \Lambda_{i}=x\sin\frac{\theta_{i}}{2}+r_{i}{\sin\theta }_{i}{+y\beta }_{i}\cos\frac{\theta_{i}}{2}\\ \Omega_{i}=x\cos\frac{\theta_{i}}{2}+r_{i}{\cos\theta }_{i}{-y\beta }_{i}\mathrm{si}n\frac{\theta_{i}}{2} \end{array}$$
(35)$$\omega =\frac{{\hat{r}}_{2}\beta_{1}\left({a+b\beta }_{1}\right)\left({a+b\beta }_{2}\right){\hat{\Omega }}_{1}{\hat{\Theta }}_{2}+{\hat{r}}_{1}{\hat{\Theta }}_{1}\left\{{\hat{r}}_{2}\left(\beta_{1}{-\beta }_{2}\right){\hat{\Theta }}_{2}-\left[\left({a+b\beta }_{1}\right)\beta_{2}\left({a+b\beta }_{2}\right){\hat{\Omega }}_{2}\right]\right\}}{{\hat{r}}_{1}{\hat{r}}_{2}\left(\beta_{1}{-\beta }_{2}\right){\hat{\Theta }}_{1}{\hat{\Theta }}_{2}}$$
(36)$${\hat{r}}_{i}{=r}_{i}\left[x_{\mathrm{Max}},0\right]\;\;\;\;\;{\hat{\Theta }}_{i}{=\Theta }_{i}\left[x_{\mathrm{Max}},0\right]\;\;\;\;{\hat{\Omega }}_{i}{=\Omega }_{i}\left[x_{\mathrm{Max}},0\right]$$

Along the notch bisector line, Equation (32) simplifies to give:(37)$$\tau_{\mathrm{xy}}=\frac{\tau_{\mathrm{xy}}^{\mathrm{Max}}\left(z\right)}{\omega }\left\{1+\frac{\left({a+b\beta }_{1}\right)\left({a+b\beta }_{2}\right)\left(\beta_{1}r_{2,0}\Theta_{1,0}\Omega_{2,0}{-r}_{1,0}\beta_{2,0}\Theta_{2,0}\Omega_{1,0}\right)}{r_{1,0}r_{2,0}\left(\beta_{1}{-\beta }_{2}\right)\Theta_{1,0}\Theta_{2,0}}\right\}$$
where:(38)$$r_{i,0}=\sqrt[{2}]{a^{2}{-x}^{2}{-b}^{2}\beta_{i}^{2}}{\;\;\;\;\Theta }_{i}{=x}^{2}{+r}_{i,0}{}^{2}{+2r}_{i,0}\cdot {x\;\;\;\;\Omega }_{i}{=x+r}_{i,0}$$

The out-of-plane stress fields, instead, can be derived starting from the solution obtained by Zappalorto and Salviato [31] for the pure antiplane shear loadings, which can be re-written as:(39)$$\begin{array}{l} \tau_{zy}=\tau_{zy}^{Max}(z)\frac{\beta_{3}^{}\sqrt{\frac{\rho }{a}}}{1-\beta_{3}^{2}\frac{\rho }{a}}\left\{\frac{r_{3}}{\sqrt{r_{31}\cdot r_{32}}}\cos\left(\theta_{3}-\frac{\theta_{31}+\theta_{32}}{2}\right)-\beta_{3}\sqrt{\frac{\rho }{a}}\right\}\\ \tau_{zx}=\tau_{zy}^{Max}(z)\frac{\beta_{3}^{2}\sqrt{\frac{\rho }{a}}}{1-\beta_{3}^{2}\frac{\rho }{a}}\left\{\frac{r_{3}}{\sqrt{r_{31}\cdot r_{32}}}\sin\left(\theta_{3}-\frac{\theta_{31}+\theta_{32}}{2}\right)\right\} \end{array}$$
where:(40)$$z_{3}=x+i\beta_{3}y=r_{3}e^{i\theta_{3}}\;\;\;z_{3}^{2}-a^{2}+\beta_{3}^{2}B^{2}=z_{3}^{2}-{\hat{C}}^{2}=\left(z_{3}^{}-{\hat{C}}^{}\right)\left(z_{3}^{}+{\hat{C}}^{}\right)$$
(41)$$\left(z_{3}^{}-{\hat{C}}^{}\right)=r_{31}e^{i\theta_{31}}\;\;\;\left(z_{3}^{}+{\hat{C}}^{}\right)=r_{32}e^{i\theta_{32}}$$
and $\tau_{zy}^{Max}(z)$ is, for a given z value, the maximum shear stress occurring at the notch tip. 

Along the notch bisector line, Equation (39) simplifies to give:(42)$$\tau_{zy}=\tau_{zy}^{Max}(z)\frac{\beta_{3}^{}\sqrt{\frac{\rho }{a}}}{1-\beta_{3}^{2}\frac{\rho }{a}}\left\{\frac{\frac{x}{a}}{\sqrt{{\left(\frac{x}{a}\right)}_{}^{2}-1+\beta_{3}^{2}\frac{\rho }{a}}}-\beta_{3}\sqrt{\frac{\rho }{a}}\right\}$$

## 4. Lateral Radiused Notch under Shear

Consider a thick plate with two lateral radiused notches with a generic notch opening angle (2α) and subjected to shear or torsion (see Figure 2). According to the previous literature (see, among the others, [33], and references reported therein), the notch boundary can be described with a hyperbola-like curve with parametric equations:(43)$$x=\frac{r_{0}}{\cos{\left(\frac{\theta }{q}\right)}^{q}}\;\;\;y=\frac{r_{0}}{\sin{\left(\frac{\theta }{q}\right)}^{q}}$$
where:(44)$$r_{0}=\frac{q-1}{q}\rho \;\;\;q=\frac{2\pi -2\alpha }{\pi }\;\;\;$$
and ρ is the notch root radius. 

The in-plane stress fields can be determined according to the solution proposed by Pastrello et al. [34]:(45)$$\begin{array}{l} \sigma_{rr}^{}=A(z)\{{\left(\frac{r_{1}}{r_{0}}\right)}_{}^{\lambda_{2}-1}\left[k_{12}\cos(1-\lambda_{2})\theta_{1}^{}-k_{11}\sin(1-\lambda_{2})\theta_{1}^{}\right]+\\ +\chi_{12}{\left(\frac{r_{1}}{r_{0}}\right)}_{}^{\mu_{2}-1}\left[k_{12}\cos(1-\mu_{2})\theta_{1}^{}-k_{11}\sin(1-\mu_{2})\theta_{1}^{}\right]+\\ +\chi_{21}{\left(\frac{r_{2}}{r_{0}}\right)}_{}^{\lambda_{2}-1}\left[k_{22}\cos(1-\lambda_{2})\theta_{2}^{}-k_{21}\sin(1-\lambda_{2})\theta_{2}^{}\right]+\\ +\chi_{22}{\left(\frac{r_{2}}{r_{0}}\right)}_{}^{\mu_{2}-1}\left[k_{22}\cos(1-\mu_{2})\theta_{2}^{}-k_{21}\sin(1-\mu_{2})\theta_{2}^{}\right]+\\ +\chi_{23}{\left(\frac{r_{2}}{r_{0}}\right)}_{}^{\zeta_{2}-1}\left[k_{22}\cos(1-\zeta_{2})\theta_{2}^{}-k_{21}\sin(1-\zeta_{2})\theta_{2}^{}\right]\} \end{array}$$
(46)$$\begin{array}{l} \sigma_{\theta \theta }^{}=A(z)\{{\left(\frac{r_{1}}{r_{0}}\right)}_{}^{\lambda_{2}-1}\left[m_{12}\cos(1-\lambda_{2})\theta_{1}^{}-m_{11}\sin(1-\lambda_{2})\theta_{1}^{}\right]+\\ +\chi_{12}{\left(\frac{r_{1}}{r_{0}}\right)}_{}^{\mu_{2}-1}\left[m_{12}\cos(1-\mu_{2})\theta_{1}^{}-m_{11}\sin(1-\mu_{2})\theta_{1}^{}\right]+\\ +\chi_{21}{\left(\frac{r_{2}}{r_{0}}\right)}_{}^{\lambda_{2}-1}\left[m_{22}\cos(1-\lambda_{2})\theta_{2}^{}-m_{21}\sin(1-\lambda_{2})\theta_{2}^{}\right]+\\ +\chi_{22}{\left(\frac{r_{2}}{r_{0}}\right)}_{}^{\mu_{2}-1}\left[m_{22}\cos(1-\mu_{2})\theta_{2}^{}-m_{21}\sin(1-\mu_{2})\theta_{2}^{}\right]+\\ +\chi_{23}{\left(\frac{r_{2}}{r_{0}}\right)}_{}^{\zeta_{2}-1}\left[m_{22}\cos(1-\zeta_{2})\theta_{2}^{}-m_{21}\sin(1-\zeta_{2})\theta_{2}^{}\right]\} \end{array}$$
(47)$$\begin{array}{l} \tau_{r\theta }^{}=A(z)\{{\left(\frac{r_{1}}{r_{0}}\right)}_{}^{\lambda_{2}-1}\left[n_{12}\cos(1-\lambda_{2})\theta_{1}^{}-n_{11}\sin(1-\lambda_{2})\theta_{1}^{}\right]+\\ +\chi_{12}{\left(\frac{r_{1}}{r_{0}}\right)}_{}^{\mu_{2}-1}\left[n_{12}\cos(1-\mu_{2})\theta_{1}^{}-n_{11}\sin(1-\mu_{2})\theta_{1}^{}\right]\\ +\chi_{21}{\left(\frac{r_{2}}{r_{0}}\right)}_{}^{\lambda_{2}-1}\left[n_{22}\cos(1-\lambda_{2})\theta_{2}^{}-n_{21}\sin(1-\lambda_{2})\theta_{2}^{}\right]+\\ +\chi_{22}{\left(\frac{r_{2}}{r_{0}}\right)}_{}^{\mu_{2}-1}\left[n_{22}\cos(1-\mu_{2})\theta_{2}^{}-n_{21}\sin(1-\mu_{2})\theta_{2}^{}\right]+\\ +\chi_{23}{\left(\frac{r_{2}}{r_{0}}\right)}_{}^{\zeta_{2}-1}\left[n_{22}\cos(1-\zeta_{2})\theta_{2}^{}-n_{21}\sin(1-\zeta_{2})\theta_{2}^{}\right]\} \end{array}$$
where, denoting with x′ and y′ the distances from the notch tip:(48)$$x_{j}=x^{\prime }+r_{0}\beta_{j}^{t_{2}}\;\;\;y_{j}=\beta_{j}^{}y^{\prime }\;\;\;r_{j}=\sqrt{x_{j}^{2}+y_{j}^{2}}\;\;\;\theta_{j}=Arg(x_{j}+iy_{j})\;\;\;j=1,2$$
and λ_2_ is a linear elastic eigenvalue to be determined by solving the following equation:(49)$$\begin{array}{l} \cos(1-\lambda_{2})\theta_{2}(\gamma )\{\;\cos(1-\lambda_{2})\theta_{1}(\gamma )\left[m_{12}(\gamma )n_{22}(\gamma )-m_{22}(\gamma )n_{12}(\gamma )\right]-\\ \sin(1-\lambda_{2})\theta_{1}(\gamma )\left[m_{11}(\gamma )n_{22}(\gamma )-m_{22}(\gamma )n_{11}(\gamma )\right]\;\}-\\ \sin(1-\lambda_{2})\theta_{2}(\gamma )\{\;\cos(1-\lambda_{2})\theta_{1}(\gamma )\left[m_{21}(\gamma )n_{12}(\gamma )-m_{12}(\gamma )n_{21}(\gamma )\right]-\\ \sin(1-\lambda_{2})\theta_{1}(\gamma )\left[m_{11}(\gamma )n_{21}(\gamma )-m_{21}(\gamma )n_{11}(\gamma )\right]\;\}=0 \end{array}$$
where γ = π − α. Parameters t_2_, μ_2_, ζ_2_, χ_12_, χ_21_, χ_22_ and χ_23_ depend on the notch geometry and the material properties and can be determined according to the procedure proposed in Ref. [34].

Moreover, one should note that parameter A(z) in Equations (45)–(47) can be expressed, for a given z value, as a function of the maximum shear stress along the notch bisector, the maximum principal stress along the notched edge or as a function of a mode 2 generalised stress intensity factor for that given plane (z value).

The out-of-plane shear stress components can be determined according to the solution proposed in Ref. [31], according to which the stress field can be written as follows:(50)$$\begin{array}{l} \tau_{z\theta }^{}=\tau_{z\theta }^{MAx}(z){\left(\frac{r_{0}\;\beta_{3}^{t_{3}}}{r_{3}^{}}\right)}_{}^{1-\lambda_{3}{}^{}}\left[\cos(1-\lambda_{3}{}^{})\theta_{3}\cos\theta_{}+\beta_{3}\sin(1-\lambda_{3}{}^{})\theta_{3}\sin\theta \right]\\ \tau_{zr}^{}=\tau_{z\theta }^{MAx}(z){\left(\frac{r_{0}\;\beta_{3}^{t_{3}}}{r_{3}^{}}\right)}_{}^{1-\lambda_{3}{}^{}}\left[\cos(1-\lambda_{3}{}^{})\theta_{3}\sin\theta_{}-\beta_{3}\sin(1-\lambda_{3}{}^{})\theta_{3}\cos\theta \right] \end{array}$$
where $\tau_{z\theta }^{MAx}(z)$ is, for a given z value, the maximum shear stress occurring at the notch tip, whereas:(51)$$x_{3}=x^{\prime }+r_{0}\beta_{3}^{t_{3}}\;\;\;y_{3}=\beta_{3}^{}y^{\prime }$$
(52)$$r_{3}=\sqrt{x_{3}^{2}+y_{3}^{2}}\;\;\;\theta_{3}=Arg(x_{3}+iy_{3})$$

In Equation (51), x′ and y′ are the distances from the notch tip in the x and y directions, respectively.

Moreover, the following equations hold valid:(53)$$t_{3}=2-\frac{Ln\frac{q-1}{q(1-\lambda_{3}{}^{})}}{Ln\beta_{3}^{}}$$
and
(54)$$\lambda_{3}=\frac{\pi }{2\left\{\mathrm{Arctan}\left[\beta_{3}\tan\gamma \right]+\pi \right\}}$$

Along the notch bisector line Equation (50) simplifies to give:(55)$$\tau_{z\theta }^{}=\tau_{z\theta }^{MAx}(z){\left(\frac{x^{\prime }}{r_{0}\beta_{3}^{t_{3}}}+1\right)}_{}^{\lambda_{3}{}^{}-1}$$

One should note that, in the case of an isotropic material, β_3_ = 1 and, accordingly, $\lambda_{3}=\frac{\pi }{2\gamma }$, according to [5,35].

## 5. Discussion and Results

In order to support the theoretical results previously described, a set of numerical simulations has been carried out. In more details, three-dimensional elastic FE analyses were performed using three different material systems, of which the elastic properties are summarised in Table 1. They represent typical properties of some polymeric composites:Material 1 represents a unidirectional carbon-fibre-reinforced epoxy laminate with the fibres oriented in the direction of the notch bisector;Material 2 represents the same material with fibres oriented in the direction normal to the notch bisector;Material 3 represents a quasi-isotropic carbon-fibre-reinforced epoxy laminate (e.g., [(0/±45/90)_n_]_S_).

Numerical investigations were carried out on thick notched discs (see Figure 3) by applying on their external surface a pure mode 2 or a pure mode 3 displacement field (see the expressions in Appendix B), in order to generate an in-plane or out-of-plane stress field in the notched solids. This modelling strategy was already used by Berto et al. [25] in order to ease the analysis of the local behaviour of a generic 3D solid.

The following notch geometries were considered:Semi-elliptical notches with notch depth a = 5 mm and different notch root radii (ρ = 0.001 mm, 0.01 mm, 0.1 mm and 1 mm); in this case, the radius of the disc was 5 mm and various thicknesses were considered (t = 1 mm, 5 mm and 10 mm);Hyperbolic notches with a notch depth a = 10 mm, a notch root radius ρ = 0.01 mm and various notch opening angles (see Table 2 and Table 3 for the stress field parameters associated with the cases analysed). 

FE analyses were carried out with the commercial FE code ANSYS, release 19.5, using 20 nodes SOLID186 brick elements with reduced integration and pure displacement formulation. 

The loads were applied as nodal displacements on the circular boundary of the disc (see Figure 3a). In more detail, the nodal coordinates of the boundaries were collected in ANSYS APDL, and, by means of an external script, the displacements to be applied were evaluated according to the expressions reported in Appendix B. Eventually, another APDL script was used to apply the nodal displacements to the disc boundary. For simulating mode 2, the symmetry of the model was exploited in order to use only one-quarter of the disc. Symmetry boundary conditions were applied to the face visible in Figure 3a, at a z-coordinate equal to half the disc thickness (i.e., the disc midplane). Anti-symmetry was applied to the nodes lying on the other symmetry plane. For simulating mode 3, only the anti-symmetry boundary conditions were used. Hence, the model needed was half of a full disc. 

An important aspect to be accounted for in the definition of the FE model parameters is the elastic property of the plate material. In fact, depending on the material orientation, the mesh requires a different number of elements to reach a convergent solution. To minimize the computational cost of the model, the number of elements and the spacing ratio, both in the in-plane and through-the-thickness directions, required a multi-variable optimization. For each model, the parameters were modified until the maximum value of $\tau_{\mathrm{zy}}^{\mathrm{Max}}\left(z\right)$ at notch tip showed a variation of less than 1%. The choice of that output variable for the convergence analysis was made under the rationale that the most critical zone for mode 2 to mode 3 coupling is near the notch tip.

Further details about the numerical validation (e.g., mesh construction for each geometry) can be found in the Appendix A retrievable online.

Initially, the attention was focused on discs with a thickness t = 1 mm loaded under pure mode 2 and weakened by elliptical notches with a root radius ρ= 0.001 mm. Results related to the plane z/t = 0.4 (where z is the distance from the mid-plane) are presented in Figure 4, Figure 5, Figure 6, Figure 7 and Figure 8. In particular, in Figure 4, the normal in-plane stress component tangent to the notch edge, σ_vv_, is plotted against the polar angle θ, measured with respect to the ellipse foci. Differently, the shear stress τ_xy_, as evaluated along the notch bisector line, is presented in Figure 5 as a function of the distance from the notch tip. As evident, in both cases, the results from the three-dimensional numerical analyses perfectly agree with the two-dimensional solution, Equations (32) and (37).

In Figure 6 and Figure 7, the attention is instead focused on out-of-plane shear stresses induced by three-dimensional effects. As evident, these stress components can be accurately predicted using the solutions derived for the antiplane deformation, Equations (39) and (42). Figure 8 makes it evident that, considering several z/t values, the induced out-of-plane shear stresses remain self-similar for various z/t values, except when very close to the free surface of the disc. Accordingly, the value of z/t = 0.4 can be regarded as representative of any z value at a sufficient distance from the free edge of the disc. The particular choice of z/t = 0.4 was made to obtain comparable magnitudes between the induced stresses and the main stresses generated by the imposed loading conditions in the numerical simulations. 

Eventually, considering different values for the disc thickness (t = 1 mm, 5 mm and 10 mm) the trend of the maximum in-plane shear stress, $\tau_{\mathrm{xy}}^{\mathrm{Max}}\left(z\right)$ and of the maximum out-of-plane shear stress $\tau_{\mathrm{zy}}^{\mathrm{Max}}\left(z\right)$ are reported in Figure 9a,b, respectively, where it can be noted that: The maximum in-plane shear stress $\tau_{\mathrm{xy}}^{\mathrm{Max}}\left(z\right)$ remains constant for most of the plane thickness, and significantly increases as approaching the free surface of the plate (Figure 9a);The maximum out-of-plane shear stress $\tau_{\mathrm{zy}}^{\mathrm{Max}}\left(z\right)$ has an almost linear trend up to z/t around 0.3. When approaching the free surface of the disc the trend becomes strongly nonlinear;Increasing the disc thickness, the intensity of the induced out-of-plane shear stresses significantly increases (Figure 9b), whereas the influence of the thickness on $\tau_{\mathrm{xy}}^{\mathrm{Max}}\left(z\right)$ is limited (Figure 9a).

Figure 10 and Figure 11 present the effects of different notch root radii in terms of out-of-plane shear stresses. In particular, it can be noted from Figure 10 that, when increasing the notch radius:The accuracy of Equation (42) decreases. Recalling the analytical treatise presented in Section 2, this is due to the terms $f^{\prime }(z)\times u(x,y)$ and $f^{\prime }(z)\times v(x,y)$, which are more and more relevant when decreasing the value of the notch root radius, when compared to g(z)⋅∂w(x,y)/∂x and g(z)⋅∂w(x,y)/∂y.The region ahead of the notch tip where the out-of-plane shear stresses are significant progressively reduces.

In addition to this, Figure 11 makes it evident that when increasing the notch radius, the intensity of $\tau_{\mathrm{zy}}^{\mathrm{Max}}\left(z\right)$ significantly reduces as well.

Subsequently, the attention was moved to discs with a thickness t = 1 mm loaded under pure mode 2 and weakened by hyperbolic notches with a root radius ρ= 0.01 mm and different notch opening angles. Results related to the plane z/t = 0.4 (where z is the distance from the mid-plane) are presented in Figure 12, Figure 13 and Figure 14. In particular, in Figure 12, the normal stress component tangent to the notch edge, σ_vv_, is plotted against the polar angle θ; also, in this case, the results from the three-dimensional numerical analyses perfectly agree with the two-dimensional solution, Equations (45)–(47). 

In Figure 13 and Figure 14, it is again proved that three-dimensional effects induce out-of-plane shear stresses that can be accurately predicted using the solutions derived for the antiplane deformation problem, Equations (50) and (55), independently of the material system considered. 

Finally, the disc made with material 2 and weakened by a hyperbolic notch with ρ= 0.01 mm and 2α = 90° was loaded under pure mode III (applying the displacement fields reported in Appendix B). For this case, Figure 15 documents the existence of induced in-plane shear stresses (induced mode 2), which can be accurately predicted using Equations (45)–(47).

## 6. Conclusions

In this paper, an analytical solution for the stress fields in the close neighbourhoods of radiused notches in thick orthotropic plates under shear loading and twisting is provided. As a first step, the equations of the three-dimensional theory of elasticity were successfully reduced to two uncoupled equations in the two-dimensional space. Later, the 3D stress field solution for orthotropic plates with radiused notches was presented, and its degree of accuracy was discussed by comparing the theoretical results and numerical data from 3D FE analyses. 

Based on the results presented and discussed, the following main conclusions can be drawn:Three-dimensional effects in thick plates or discs induce coupling phenomena between loading modes. In particular, out-of-plane shear stresses (mode 3) are induced on in-plane shear-loaded (mode 2) solids and can be accurately predicted using the solutions derived for the antiplane deformation problem. The vice versa is also true, independently of the orthotropic material system considered.Increasing the disc or plate thickness results in an increase in the intensity of stresses induced by 3D effects;The intensity of induced stresses is also significantly affected by the notch root radius, and the phenomenon tends to become negligible in the presence of large notch root radii.

## Figures and Tables

**Figure 1 polymers-15-02013-f001:**
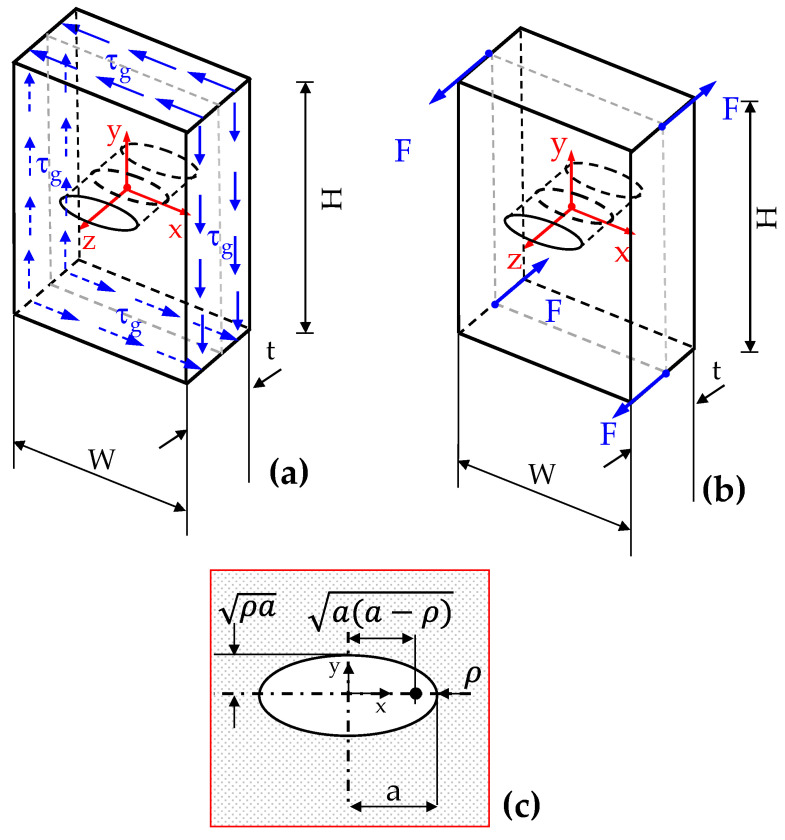
Elliptic hole in a three-dimensional plate under in-plane shear (**a**) or twisting (**b**). In (**c**) a schematic of elliptic hole is reported together with the dimensions used in the mathematical treatise and in the results’ discussion.

**Figure 2 polymers-15-02013-f002:**
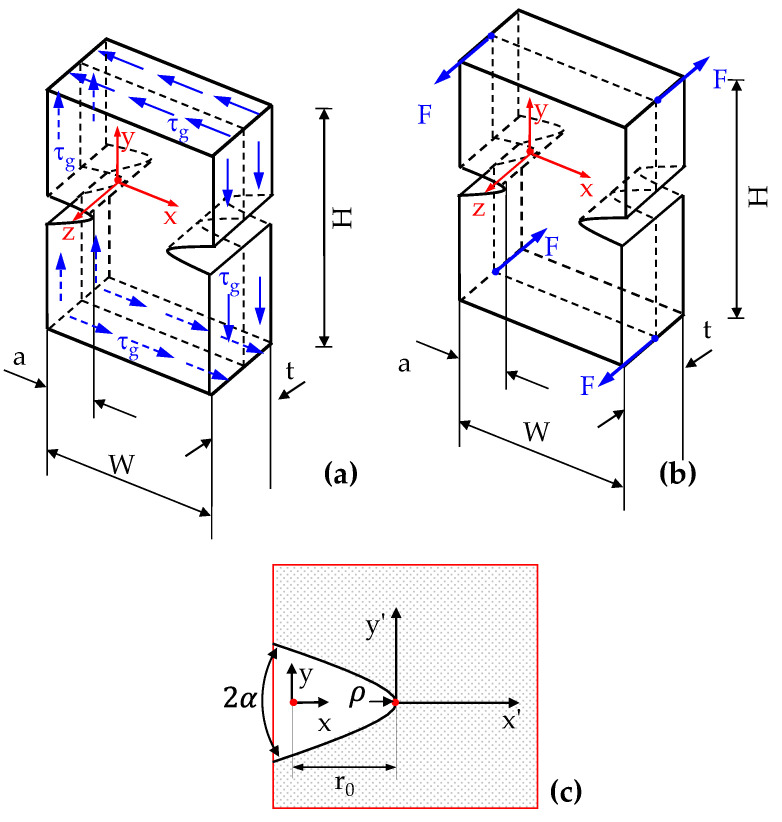
Rounded V-shaped (hyperbolic) notches in a three-dimensional plate under in-plane shear (**a**) or twisting (**b**). In (**c**) a schematic of elliptic hole is reported together with the dimensions used in the mathematical treatise and in the results’ discussion.

**Figure 3 polymers-15-02013-f003:**
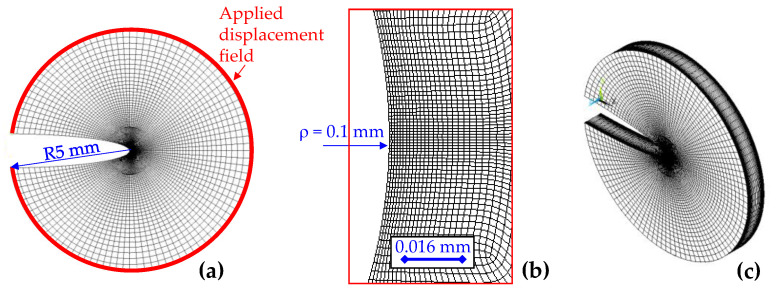
An example of the FE mesh used for an elliptic notch of ρ= 0.1 mm, a = 5 mm and t = 1 mm. All the discs used for the simulations of elliptic notches have a radius of 5 mm, while those for hyperbolic notches are 10 mm. In (**a**), a side view of the notched disc is shown, highlighting in red the border where the displacement field is applied. In (**b**), a close-up zoom of the mesh at the notch tip is reported. In (**c**), an isometric view of the meshed geometry is shown.

**Figure 4 polymers-15-02013-f004:**
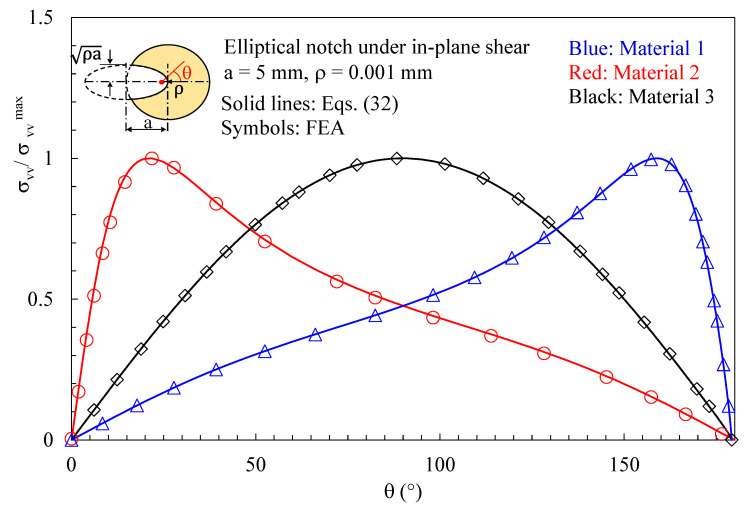
Disc with t = 1 mm and weakened by an elliptical notch with ρ= 0.001 mm. Mode 2 loadings, different materials. In-plane stress component σ_vv_ along the notched edge together with a comparison with Equation (32). Distance from the midplane z/t = 0.4.

**Figure 5 polymers-15-02013-f005:**
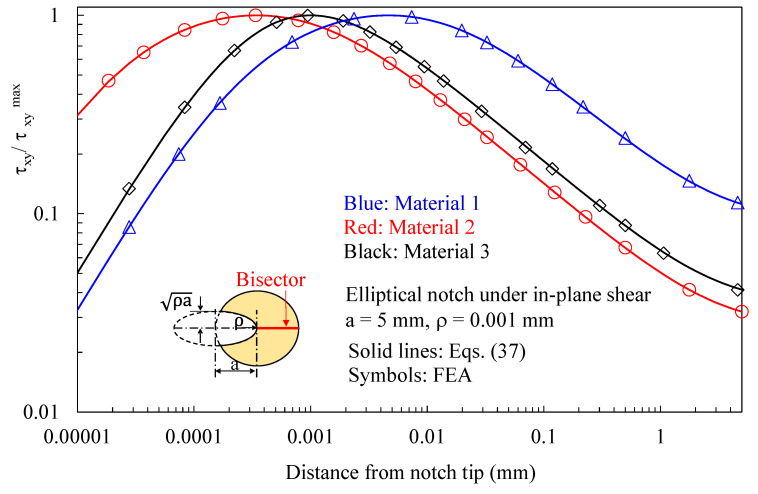
Disc with t = 1 mm and weakened by an elliptical notch with ρ= 0.001 mm. Mode 2 loadings, different materials. In-plane stress component τ_xy_ along the notch bisector together with a comparison with Equation (37). Distance from the midplane z/t = 0.4.

**Figure 6 polymers-15-02013-f006:**
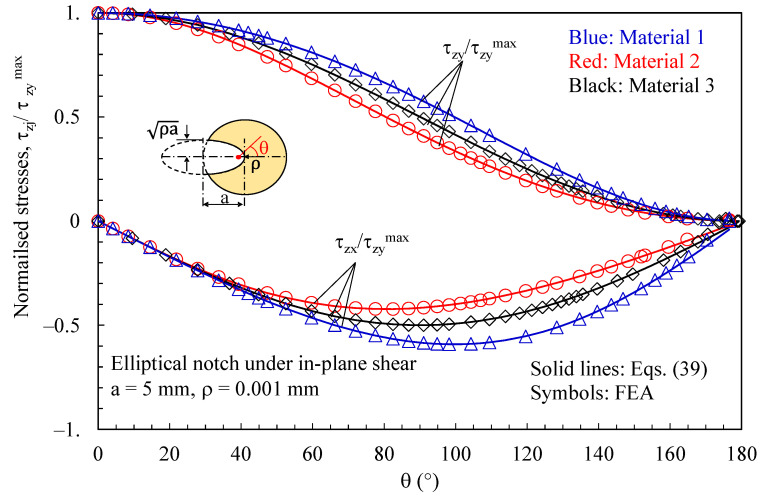
Disc with t = 1 mm and weakened by an elliptical notch with ρ= 0.001 mm. Mode 2 loadings, different materials. Induced out-of-plane stress components τ_xz_ and τ_yz_ along the notched edge together with a comparison with Equation (39). Distance from the midplane z/t = 0.4.

**Figure 7 polymers-15-02013-f007:**
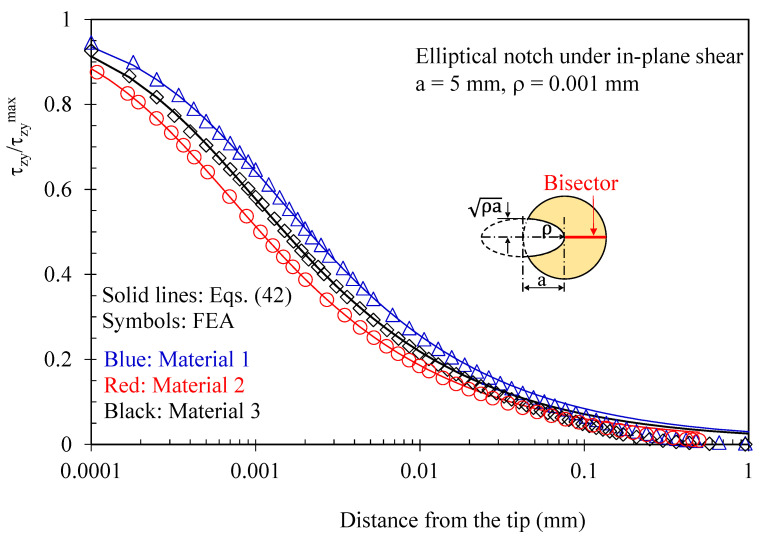
Disc with t = 1 mm and weakened by an elliptical notch with ρ= 0.001 mm. Mode 2 loadings, different materials. Induced out-of-plane stress component τ_zy_ along the notch bisector together with a comparison with Equation (42). Distance from the midplane z/t = 0.4.

**Figure 8 polymers-15-02013-f008:**
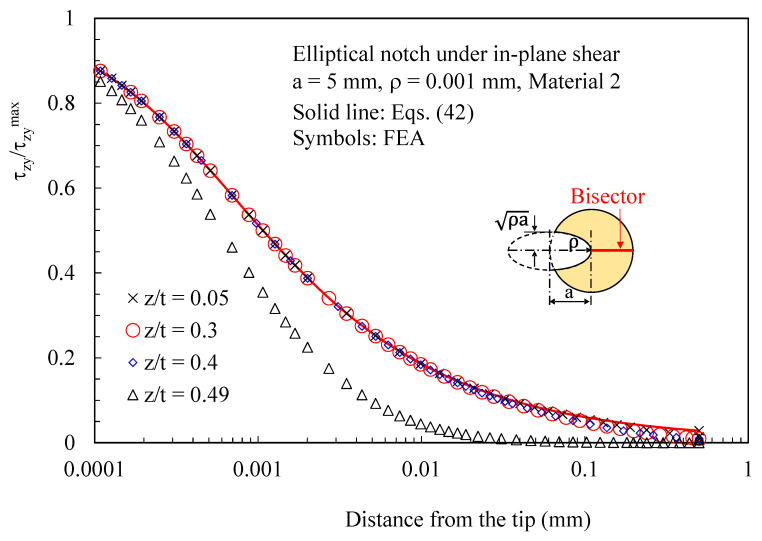
Disc with t = 1 mm and weakened by an elliptical notch with ρ= 0.001 mm. Mode 2 loadings, material 2. Induced out-of-plane stress component τ_zy_ along the notch bisector together with a comparison with Equation (42). Different distances from the midplane (z/t).

**Figure 9 polymers-15-02013-f009:**
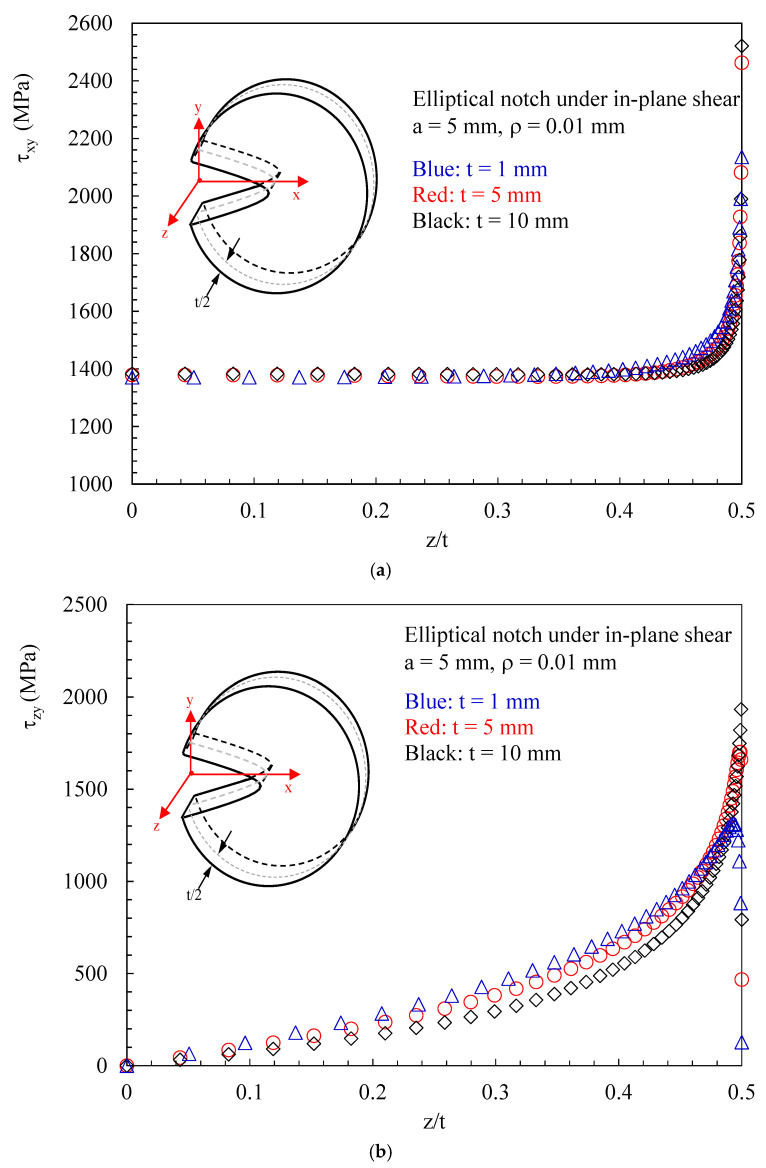
Disc weakened by an elliptical notch with ρ= 0.01 mm. Mode 2 loadings, material 2. $\tau_{\mathrm{xy}}^{\mathrm{Max}}\left(z\right)$ (**a**) and $\tau_{\mathrm{zy}}^{\mathrm{Max}}\left(z\right)$ (**b**) as a function of the distance from the mid-plane (z/t). Different thicknesses (t = 1 mm, 5 mm and 10 mm).

**Figure 10 polymers-15-02013-f010:**
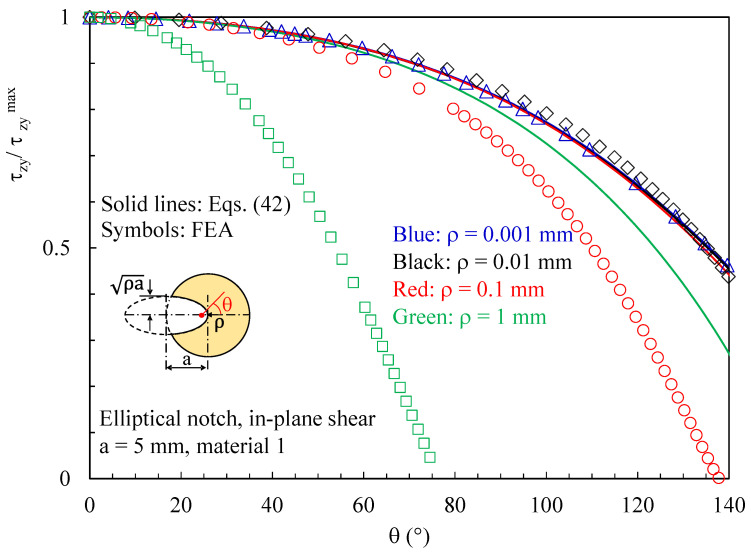
Discs with t = 1 mm and weakened by elliptical notches. Mode 2 loadings, material 2. Induced out-of-plane stress component τzy along the notch bisector together with a comparison with Equation (42). Different values for the notch root radius.

**Figure 11 polymers-15-02013-f011:**
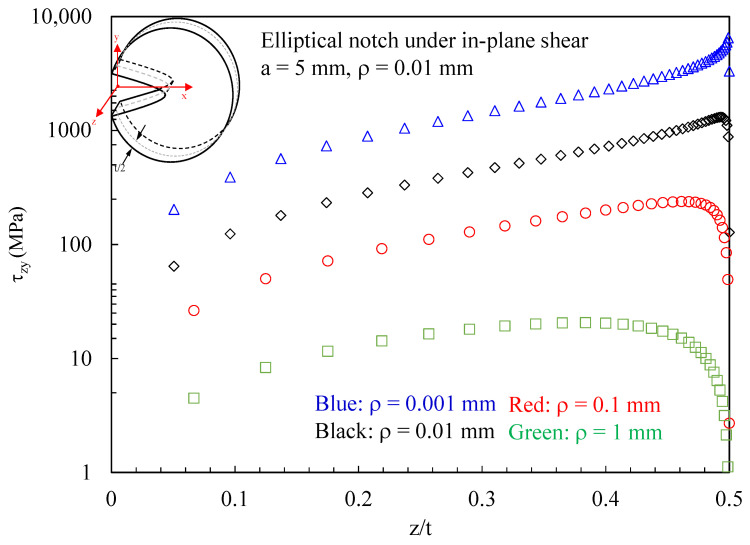
Discs with t = 1 mm and weakened by elliptical notches. Mode 2 loadings, material 1. $\tau_{\mathrm{zy}}^{\mathrm{Max}}\left(z\right)$ as a function of the distance from the mid-plane (z/t). Different values for the notch root radius.

**Figure 12 polymers-15-02013-f012:**
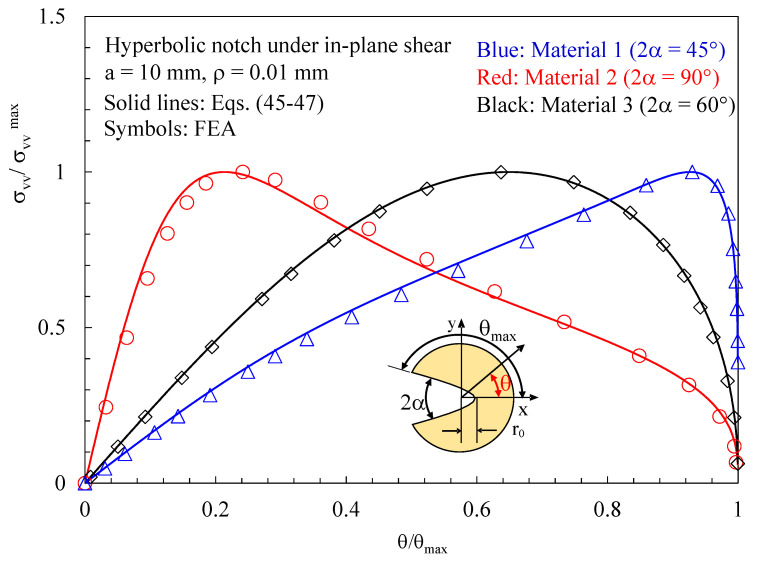
Discs with t = 1 mm and weakened by hyperbolic notches with ρ= 0.01 mm. Mode 2 loadings, different materials. In-plane stress component σ_vv_ along the notch edge together with a comparison with Equations (45)–(47). Distance from the midplane z/t = 0.4.

**Figure 13 polymers-15-02013-f013:**
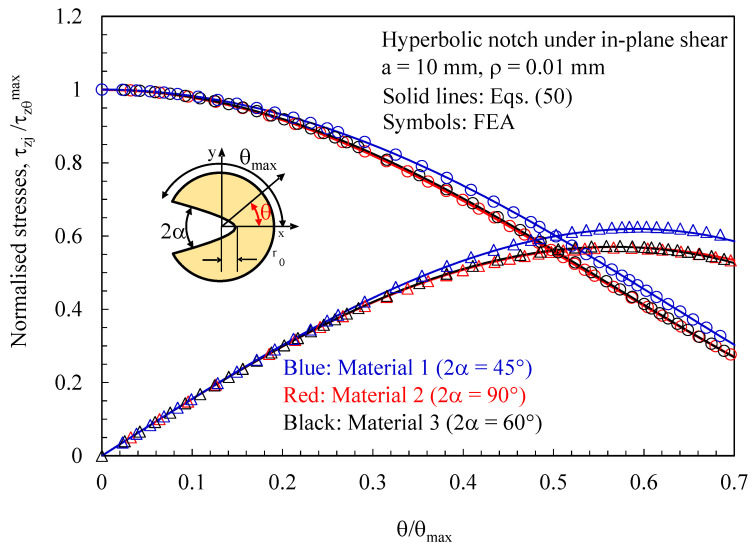
Discs with t = 1 mm and weakened by hyperbolic notches with ρ= 0.01 mm. Mode 2 loadings, different materials. Induced out-of-plane stress components τ_xz_ and τ_yz_ along the notched edge together with a comparison with Equation (50). Distance from the midplane z/t = 0.4.

**Figure 14 polymers-15-02013-f014:**
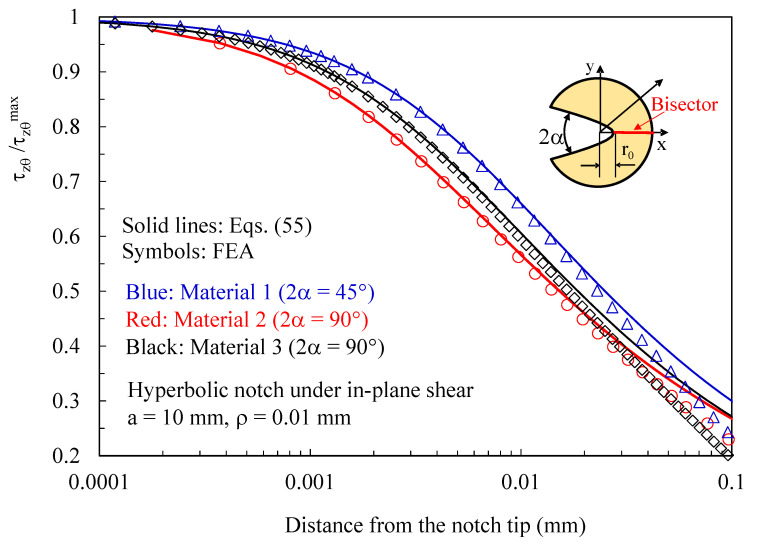
Discs with t = 1 mm and weakened by hyperbolic notches with ρ= 0.01 mm. Mode 2 loadings, different materials. Induced out-of-plane stress component τ_zy_ along the notch bisector together with a comparison with Equation (55). Distance from the midplane z/t = 0.4.

**Figure 15 polymers-15-02013-f015:**
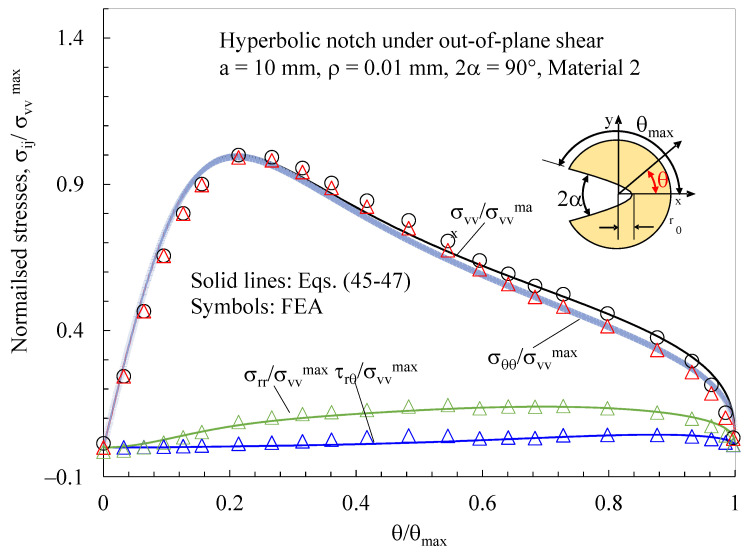
Discs with t = 1 mm and weakened by hyperbolic notches with ρ= 0.01 mm and 2α = 90°. Mode 3 loadings, materials 2. Induced in-plane stress components along the notched edge together with a comparison with Equations (45)–(47). Distance from the midplane z/t = 0.4.

**Table 1 polymers-15-02013-t001:** Properties of the materials used in the numerical analyses.

	E_x_ (GPa)	E_y_ (GPa)	E_z_ (GPa)	ν_xy_	ν_xz_	ν_yz_	G_xy_ (GPa)	G_xz_ (GPa)	G_yz_ (GPa)	β_1_	β_2_	β_3_
Material 1	160	10	10	0.3	0.3	0.4	5	5	3.57	0.6614	5.5587	1.1835
Material 2	10	160	10	0.01875	0.4	0.3	5	3.57	5	0.1798	1.5120	0.8450
Material 3	70	70	70	0.3	0.3	0.3	26.9	26.9	26.9	0.9993	1.0006	1.0000

**Table 2 polymers-15-02013-t002:** In-plane stress field parameters for the hyperbolic notches used in the numerical analyses.

	t_2_	λ_2_	μ_2_	ζ_2_	χ_12_	χ_21_	χ_22_	χ_23_
Material 1, 2α = 45°	1.6849	0.8757	0.3533	−0.3806	0.2997	−0.2733	0.0888	−0.4661
Material 2, 2α = 90°	1.7319	0.7788	0.2551	−0.8948	0.1629	−0.4744	0.0152	−0.0417
Material 3, 2α = 60°	1.5200	0.7309	0.1924	−3.7178	0.0738	−0.9996	−0.0734	−4.3·10^−07^

**Table 3 polymers-15-02013-t003:** Out-of-plane stress field parameters for the hyperbolic notches used in the numerical analyses.

	t_3_	λ_3_	q
Material 1, 2α = 45°	1.8111	0.5848	1.7500
Material 2, 2α = 90°	1.6055	0.6438	1.5000
Material 3, 2α = 60°	1.7519	0.6000	1.6667

## Data Availability

The data that support the findings of this study are available on request from the corresponding author. Yet, in the Appendix A the procedure and the scripts for replicating the FE analyses are available.

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
