# Peer review of "Three-Dimensional Stress Fields in Thick Orthotropic Plates with Sharply Curved Notches under In-Plane and Out-of-Plane Shear"

_polymers, 2023, doi:10.3390/polym15092013_

Round 1

Reviewer 1 Report

This manuscript provides an analytical solution for the stress fields in the close neighborhoods of radiused notches in thick orthotropic plates under shear loading and twisting. The approximate 3D solution derived for pointed notches is extended to orthotropic plates with holes or lateral radiused notches with smaller tip radius and the existence of coupled modes for these plates is documented.

While the investigation on 3D stress distribution in an orthotropic plate with pointed notches or holes is not new since some studies have been reported, the plates with radiused notches in this study shows some interests. The following concerns should be carefully addressed before the manuscript is further considered.

1. The authors did not clearly indicate whether the research object is a notched rectangular plate or a notched disk in Figure 4, Figure 5 and other places.

2. Many stress component distribution results are only given on the plane z/t=0.4 (where z is the distance from the mid plane, t is the plate thickness). Considering that the research object is thick plate, the reason why only this plane is selected should be explained.

3. The authors mentioned that Figure 11 made it evident that increasing the notch radius, the intensity of maximum out-of-plane stress component tau_zyMax(z) significantly reduced, but when the radius increases from 0.01mm to 0.1mm, tau_zyMax(z) does not decrease, which should be explained.

4. While the authors gave the stress field distribution with polar angle changing from 0° to 180° when studying the disk with elliptical notch, only the stress field distribution with polar angle changing from 0° to 120° was given when the notch became hyperbolic. This difference should be explained.

5. Some important relevant studies should be reviewed in the revised manuscript such that the reader may have a comprehensive view of the recent progress in the field, e.g., Shi et al., Journal of Applied Mechanics, 2020, 87, 021004; Wang et al., Journal of Applied Mechanics, 2018, 85, 071009.

Reviewer 2 Report

The paper is good, but I have a few comments that should be clarified by the authors - some of them decided that I did not give the final grade as "minor revision", but "major revision".

The manuscript should include nomenclature - a list of symbols and all markings - in order not to disturb the layout of the MDPI Publisher, it may be placed at the end of the paper - some publishers do so.

In relation to scientific papers, I recommend using expressions such as "paper, scientific article, manuscript, scientific thesis etc" - I do not recommend using the term "work".

In the manucript, the authors showed 35 literature items - of which in 17 one of the authors is a co-author. The MDPI Editor should check for appropriate self-citations by authors. If the authors supplement the paper with other publications on the subject of the paper - they will significantly expand the list of literature items, they can leave the current positions - but it cannot turn out that they make their publication an option to obtain further citations to the SCOPUS or WEB OF SCIENCE database.

Formula (1a) is missing the "=" sign.

I propose to introduce additional markings (a), (b) and (c) in Figures 1 and 2 - sign the following indications in the figure.

The following part of the article is perfectly presented mathematically - the authors have illustrated their skills, derived the appropriate formulas, which they intended to verify numerically. I wonder if part of the mathematical analysis should not be transferred to the supplement - only at the beginning indicate the starting equations, boundary conditions, a short way of reasoning in the form of a block diagram (several additional figure required) and the final result - both in the form of mathematical formulas and illustrations in the form of appropriate graphs - theoretical distributions of subsequent components of the stress tensor, strain tensor or displacement vector.

In the next step, I would order numerical verification of the obtained solution.

As regards the description of the numerical model, the authors provided too little information. The authors directed their considerations towards the Boundary Layer Approach – later named Modify Boundary Layer Approach. The paper lacks the dimensions of the modeled disk - I require the radius of the disk, its thickness, specific dimensions of the notches. If the authors do not provide the dimensions of the disc, they should show that the external dimensions of the disc do not affect the distribution of mechanical fields in the vicinity of the notch tip. The description shows that the authors test two types of notches - please complete the paper with specific figures of the disk with notches and dimensions - preferably technical drawings. In the case of a mesh of finite elements, in addition to the type of elements (20 nodes brick elements), please provide the number of points of numerical integration in the element, please provide information whether the results provided by the authors are given in nodes or Jacobian points. It is required in the paper to present the appearance of the finite element mesh in the vicinity of the notch tip and in the vicinity of the notch edges. Please provide the dimensions of the finite element (information like "with a very fine mesh just to assure the accuracy of the numerical results in the highly stressed regions, with elements close to the notch tip were chosen to be at least 25 times smaller than the value of the tip radius." is too sparse). In the description of the numerical model, I am asking for information about the total number of nodes and finite elements in the model. Specific information about the density of the finite element mesh in the radial and angular directions must be provided. This should also be illustrated with a proper figure - division the whole disk or half or a quarter with applied boundary conditions and load, enlargement of a fragment of the finite element mesh in the vicinity of the notch tip and in the vicinity of the entire notch We also do not know what the division of the disk was in the direction of thickness - how many layers of finite elements were assumed by the authors in the direction of the thickness of the disk, what was the level of density of finite elements in the thickness direction In the axis of the disc - the disc, the stress gradient does not change rapidly, while at the edge the changes are very clear and rapid. This must necessarily be explained. We also do not know what the authors modeled - whether they used the axes of symmetry existing in the disc.

What can the authors say about the convergence of the solution in the numerical model? Has the convergence of the numerical model been tested in terms of division into elements terminated in front of the notch tip and in the direction of thickness?

The presentation of the results should show the influence of notch dimensions on the distribution of full mechanical fields (stresses, strains and displacements), the influence of thickness on the distribution of these fields, the influence of material characteristics. Graphs should be prepared in physical and normalized coordinates for the radial distance from the notch tip and for a densely divided angular direction (radial - from 0 to 180 degrees). In addition, it is worth showing changes in the mechanical fields along the axis of the specimen, at the edge of the specimen in the intermediate layers.

Some of these recommendations can be ignored by the authors - there are such graphs in the manuscript, but some need to be supplemented in order for the paper to be complete.

In addition, the paper is interesting and worth recommending.

Please correct the paper and resubmit for review.

Round 2

Reviewer 2 Report

The authors, with their comments and responses to my comments, convinced me of the importance of this manuscript.

The reviewed article is really very good. The contribution of the authors to its creation should be appreciated.

The authors took into account all my corrections. I accept their responses to my comments.

I recommend the manuscript for publication.